# Molecular basis of fatty acid taste in *Drosophila*

Ji-Eun Ahn, Yan Chen, Hubert Amrein*

Department of Molecular and Cellular Medicine, Health Science Center, Texas A&M University , College Station, Texas, United States

**Abstract** Behavioral studies have established that *Drosophila* appetitive taste responses towards fatty acids are mediated by sweet sensing Gustatory Receptor Neurons (GRNs). Here we show that sweet GRN activation requires the function of the *Ionotropic Receptor* genes *IR25a*, *IR76b* and *IR56d*. The former two *IR* genes are expressed in several neurons per sensillum, while *IR56d* expression is restricted to sweet GRNs. Importantly, loss of appetitive behavioral responses to fatty acids in *IR25a* and *IR76b* mutant flies can be completely rescued by expression of respective transgenes in sweet GRNs. Interestingly, appetitive behavioral responses of wild type flies to hexanoic acid reach a plateau at ~1%, but decrease with higher concentration, a property mediated through IR25a/IR76b independent activation of bitter GRNs. With our previous report on sour taste, our studies suggest that IR-based receptors mediate different taste qualities through cell-type specific IR subunits.
DOI: https://doi.org/10.7554/eLife.30115.001

*For correspondence:
amrein@tamhsc.edu

**Competing interests:** The authors declare that no competing interests exist.

## Introduction

Detection of food compounds plays a critical role in feeding behavior. Likewise, the ability to avoid harmful chemicals is essential to navigate the evaluation of suboptimal food sources that might contain toxins or are contaminated with microorganism producing harmful chemicals. Hence, animals have evolved chemoreceptors to detect and discriminate between nutritious food chemicals, and chemoreceptors that sense non-nutritious and potentially harmful chemicals. Receptors for sugars, amino acids and fatty acids have been identified in mammals (*Cartoni et al., 2010*; *Galindo et al., 2012*; *Max et al., 2001*; *Montmayeur et al., 2001*; *Nelson et al., 2001*; *Zhao et al., 2003*), and other vertebrates (*Ishimaru et al., 2005*), albeit in some, receptor types for certain nutritious chemicals have been lost (*Baldwin et al., 2014*; *Jiang et al., 2012*; *Jin et al., 2011*; *Lagerström et al., 2006*; *Zhao et al., 2012*). In mice, these various types of receptors are expressed in distinct populations of taste cells located mainly in taste buds on the tongue. For example, three different groups of cells respond to calorie-rich carbohydrates, amino acids and fatty acids, respectively, while a separate group of cells is dedicated to the detection of salt (NaCl), a modest amount of which is essential for many cellular functions. Lastly, two functionally distinct types of taste cells detect diverse organic, but repugnant chemicals (phenols, alkaloids etc.) and acids, perceived as bitter and sour, respectively (*Liman et al., 2014*). Thus, the intrinsic quality of different taste chemicals is initially encoded in the taste periphery, which is transmitted through dedicated sensory pathways, a concept generally referred as the labeled line hypothesis of taste coding (*Dethier, 1978*; *Di Lorenzo, 2000*; *Spector and Travers, 2005*).

With the exception of sugars, perception of nutritious chemicals is relatively poorly understood in insects. In *Drosophila*, food and other soluble chemicals are detected by taste sensilla (comparable to mammalian taste buds) located on the labial palps and the legs. Most of these sensilla harbor four Gustatory Receptor Neurons (GRNs), each of which is thought to mediate a taste quality: sweet, water, low salt, and bitter/high salt (*Liman et al., 2014*; *Vosshall and Stocker, 2007*). The only

appetitive taste modality that has been studied in detail is that of sweet taste, elicited by a relative small group of dietary sugars, such as glucose, sucrose, trehalose, maltose, melizitose and raffinose, found mostly in fruits. Genetic studies, combined with behavioral analysis, electrophysiology and Ca²⁺ imaging have revealed that a group of eight sugar *Gustatory receptor* genes (*Gr5a*, *Gr61a* and *Gr64a-f*) is mostly responsible for the detection of sugars (*Dahanukar et al., 2007*; *Fujii et al., 2015*; *Jiao et al., 2008*; *Miyamoto et al., 2013*; *Slone et al., 2007*; *Yavuz et al., 2014*). With the exception of *Gr5a*, all sugar *Gr* genes are expressed at most in a single GRN per sensillum (*Fujii et al., 2015*; *Slone et al., 2007*), which is generally referred to as the sweet GRN. Indeed, electrophysiological studies on a small number of labellar taste sensilla (*Dahanukar et al., 2007*) and both Ca²⁺ imaging and electrophysiological recordings on tarsal sensilla of the most distal segment of the forelegs (*Ling et al., 2014*; *Miyamoto et al., 2013*; *Yavuz et al., 2014*) have confirmed that sweet GRNs respond specifically to sugars, but not to bitter compounds or salt. Interestingly, sweet GRNs vary in the number of expressed sugar *Gr* genes (*Fujii et al., 2015*), providing different GRNs with the potential for distinct sugar sensing specificities.

In contrast to sweet taste, little is known about the cellular and molecular basis of amino acid and fatty acid taste in insects. While both these nutrients are essential for growth and development during larval life, their relevance in adults is mainly restricted to females, which require fat and protein for the production of eggs. Evidence for appetitive taste of fatty acids in *Drosophila* has been demonstrated using the classical Proboscis Extension Reflex (PER) assay (*Masek and Keene, 2013*), but whether flies can sense amino acids through their taste sensory system is less clear and appears at least in part to depend on the internal nutrient status (*Toshima and Tanimura, 2012*). Regardless, no defined set of taste neurons that respond to amino acids have been described to date.

Here, we employed a genetic approach to investigate the cellular and molecular basis for fatty acid taste. Using Ca²⁺ imaging, we report that tarsal sweet GRNs are activated by fatty acids, and we show that IR25a and IR76b are expressed in numerous GRNs, including sweet GRNs, where they are required for their activation by fatty acids, but not by sugars. RNAi knock-down of a third *IR* transcript, *IR56d*, in sweet GRNs also abolished fatty acid responses both at the behavioral as well as the cellular level. In contrast to IR25a and IR76b, which are also required for sour taste and expressed in respective acid sensing GRNs (*Chen and Amrein, 2017*), expression of IR56d is confined to sweet GRNs, suggesting a more restricted role for this receptor in fatty acid taste.

In contrast to responses to sugars, which increase in a concentration dependent manner to reach a plateau, we observed that wild type flies show maximal PER responses at modest concentrations to one fatty acids, (hexanoic acid), while higher concentrations led to a reduction in PER responses. Interestingly, we found that hexanoic acid also activates bitter GRNs, albeit in an *IR25a/IR76b* independent manner, suggesting that bitter and sweet GRN activation modulate feeding response to this ligand. Consistent with this hypothesis, inhibition of neural transmission of sweet GRNs abolishes PER responses completely, while the same manipulation of bitter GRNs results in further increase in PER responses to hexanoic acid. These observations suggest a model in which activation of bitter GRNs counteract the activation of sweet GRNs to suppress feeding responses to hexanoic acid, possibly to moderate intake of this nutrient compound.

## Results

The notion that sweet GRNs are exclusively tuned to sugar has recently been challenged, as it was shown that flies exhibit sweet GRN -dependent appetitive behavioral responses to fatty acids (*Masek and Keene, 2013*). Thus, we first wanted to confirm these findings using flies with intact and functionally impaired, tetanus toxin (TNT) expressing sweet GRNs, respectively (*Figure 1*). Both sets of control flies (*Gr64f-GAL4/+* and *+ /UAS-TNT*) showed robust PER responses to fatty acids and the sugar sucrose when tarsi were stimulated, while flies with inactivated sweet GRNs (*Gr64f-GAL4/UAS-TNT*) lost PER responses to all these food chemicals (*Figure 1A*). We note that somewhat lower PER responses were elicited when the labellum was stimulated with fatty acids (*Figure 1—figure supplement 1*). To explore whether fatty acid taste requires the sugar *Gr* genes, we used a sugar *Gr* deficient fly strain (*Gr5aᴸᵉˣᴬ*; *ΔGr61a ΔGr64a-f*) (*Fujii et al., 2015*; *Yavuz et al., 2014*) and tested PER responses to fatty acids and sucrose (*Figure 1B*). While PER responses to fatty acids were unaffected or higher in these octuple mutant flies, they were severely reduced to sucrose (*Figure 1B*), indicating that other receptors must be expressed in sweet GRNs to detect fatty acids. Use of different taste

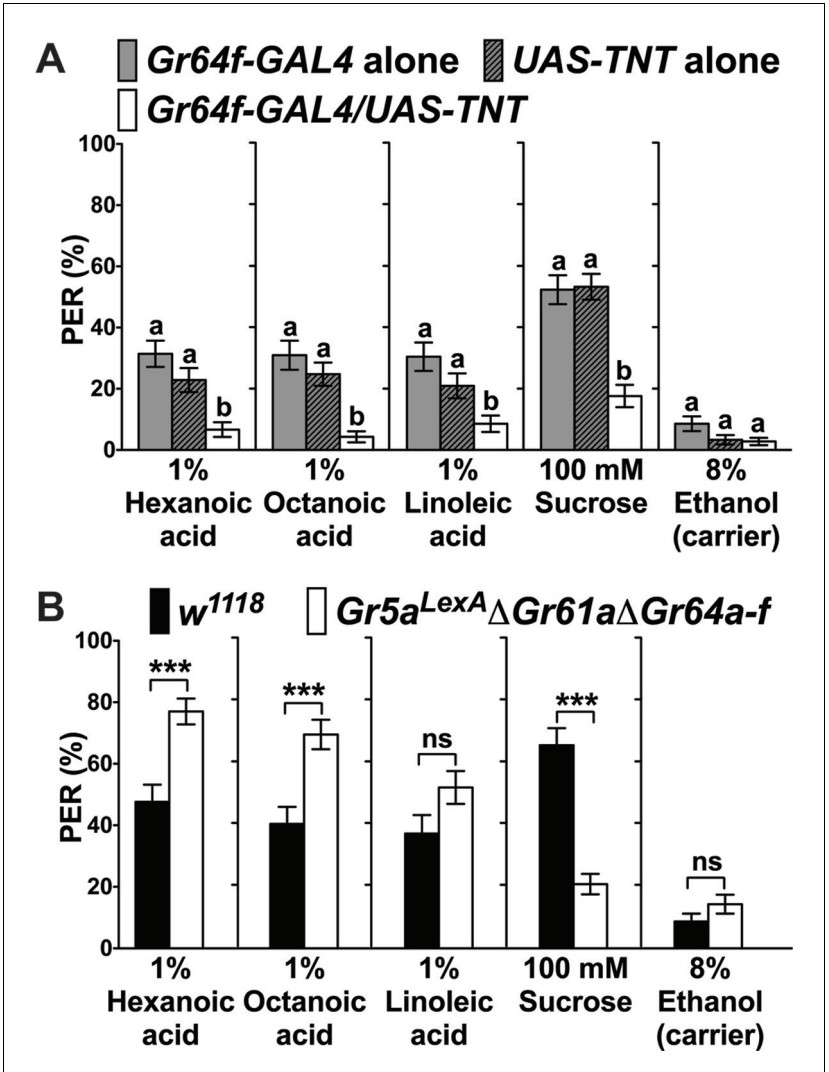

**Figure 1.** Sweet GRNs, but not sugar *Gr* genes, are necessary for fatty acid taste. (**A**) Sweet GRNs are necessary for fatty acid sensing. Inactivation of sweet GRNs (*Gr64f-GAL4/UAS-TNT*) leads to loss of PER responses to both sucrose and fatty acids. Control flies (*Gr64f-GAL4/+* and *UAS-TNT/+*) show robust PER responses to fatty acids and sucrose. Each bar represents the mean ± SEM of PER responses (n = 60–70 flies). Bars with different letters are significantly different (Kruskal-Wallis test by ranks with Dunn's multiple comparison tests, p<0.05). Each y-axis delineates groups for Kruskal-Wallis test. (**B**) Sugar *Gr* genes are dispensable for fatty acid sensing. Octuple mutant flies lacking all sugar *Gr* genes (*Gr5a^LexA^;ΔGr61aΔGr64a-f*) exhibit PER responses to fatty acids similar to flies with functional sugar *Gr* genes (*w^1118^*), but they have severely reduced PER response to sucrose. The residual PER responses to sucrose of octuple mutant flies is mediated by *Gr43a* (***Miyamoto et al., 2012***). Each bar represents the mean ± SEM of PER responses (n = 42–83 flies). Asterisks indicate a significant difference between the mutant and control flies (Two-tailed, Mann-Whitney U test, ***p<0.001, ns: not significant). Each y-axis delineates groups for Mann-Whitney U test. The genotype of octuple mutant flies is *R1 Gr5a^LexA^; +; ΔGr61a ΔGr64a-f* (***Yavuz et al., 2014***). Source data for summary graphs are provided in ***Figure 1—source data 1***.
DOI: https://doi.org/10.7554/eLife.30115.002

The following source data and figure supplements are available for figure 1:

**Source data 1.** PER responses to fatty acids of flies with impaired neurons and genes.
DOI: https://doi.org/10.7554/eLife.30115.007

**Figure supplement 1.** Flies show robust PER responses when the leg is stimulated, but is weaker when the labial palps are stimulated.
DOI: https://doi.org/10.7554/eLife.30115.003

*Figure 1 continued on next page*

*Figure 1 continued*

**Figure supplement 1—source data 1.** PER responses to fatty acids upon stimulation of legs and labellum of wild-type flies ($w^{1118}$) .
DOI: https://doi.org/10.7554/eLife.30115.004

**Figure supplement 2.** PLC signaling is required for sweet GRN responses to fatty acids.
DOI: https://doi.org/10.7554/eLife.30115.005

**Figure supplement 2—source data 1.** $Ca^{2+}$ responses of sweet GRNs associated with 5b (A), 5s (B) or 5v (C) sensilla of *norpA* mutant flies to fatty acids.
DOI: https://doi.org/10.7554/eLife.30115.006

receptors for mediating the two taste modalities is consistent with the finding that mutations in *norpA*, which encodes a phospholipase C, affected PER responses to fatty acids, but not to sugars (*Masek and Keene, 2013*). Moreover, and consistent with this behavioral phenotype, sweet taste neurons of *norpA* mutant flies had strongly reduced responses to fatty acids, but not sucrose, using a $Ca^{2+}$ imaging assay (*Figure 1—figure supplement 2*).

## Broadly expressed *IR25a* and *IR76b* are required for PER responses to fatty acids

In mammals, fatty acid taste is mediated by at least two G-protein coupled receptors, GPR40 and GPR120 (*Cartoni et al., 2010*; *Matsumura et al., 2007*). However, BLAST searchers showed that no homologs for these genes are found in the *Drosophila* genome. To identity putative candidates for *Drosophila* fatty acid taste receptors, we turned our attention to members of the *Ionotropic Receptor* (*IR*) gene family. *IRs* are derived from *ionotropic Glutamate Receptors* (*iGluRs*) genes and expanded in number during arthropod evolution, comprising a family of 61 genes in *Drosophila* (*Benton et al., 2009*; *Croset et al., 2010*). The role of IR proteins has been extensively characterized in the olfactory system, where they form multimeric receptors for the detection of an array of different odorant molecules (*Abuin et al., 2011*; *Ai et al., 2013*; *Benton et al., 2009*; *Hussain et al., 2016*; *Silbering et al., 2011*), but IR based receptors have also been implicated in temperature sensing (*Knecht et al., 2016*; *Ni et al., 2016*), humidity sensing (*Enjin et al., 2016*; *Knecht et al., 2017*; *Knecht et al., 2016*) and taste perception (*Croset et al., 2016*; *Ganguly et al., 2017*; *Hussain et al., 2016*; *Koh et al., 2014*; *Zhang et al., 2013*). Of particular interest were *IR25a* and *IR76b*, because these two genes are enriched in taste neurons (*Cameron et al., 2010*). We therefore examined the expression of *IR-GAL4/QF* drivers for both genes and found that they were indeed expressed in multiple GRNs per sensillum (*Figure 2*). In tarsal sensilla, their expression largely overlapped (*Figure 2F*) and included the sweet and bitter GRNs that express sugar and bitter *Gr* genes, respectively (*Figure 2B–E*), thus identifying them as possible subunits of fatty acid taste receptors. Both genes were also expressed in multiple neurons per sensillum in the labial palps, albeit co-expression was incomplete in this taste organ (*Figure 2—figure supplement 1*). To examine a possible role for *IR25a* and *IR76b* in fatty acid taste, we conducted PER assays with flies from each mutant strain to hexanoic, octanoic and linoleic acids and found that PER responses were abolished in both *IR25a^{-/-}* and *IR76b^{-/-}* flies, a phenotype completely (hexanoic acid and octanoic acid) or partially (linoleic acid) rescued when expression was restored with *UAS-IR25a* or *UAS-IR76b* transgenes (*Figure 3A and B*). Additionally, *IR25a^{-/-}* and *IR76b^{-/-}* flies showed normal PER responses to sucrose, indicating that the role of these two genes is specific to fatty acid taste and not required for sweet taste to sugars. Importantly, when IR25a function was provided in sweet GRNs of *IR25a^{-/-}* flies, their PER responses were restored to levels indistinguishable from controls (*Figure 3C*). Likewise, expression of *IR76b* in sweet GRNs alone also rescued PER responses to fatty acids in *IR76b^{-/-}* flies (*Figure 3D*). Together, these observations indicate that the functions of IR25a and IR76b are required in sweet GRNs for robust PER responses to fatty acids, but not to sugars.

## Sweet neurons require *IR25a* and *IR76b* for fatty acid sensing

To obtain direct evidence that *IR25a* and *IR76b* mediate fatty acid sensing in taste neurons, we performed $Ca^{2+}$ imaging experiments on tarsal sensilla preparations (*Miyamoto et al., 2013*, *2012*) of wild type control flies ($w^{1118}$), *IR25a^{-/-}* and *IR76b^{-/-}* flies that expressed the calcium sensor GCaMP6m in sweet GRNs (*Dahanukar et al., 2007*; *Fujii et al., 2015*) (*Figure 4*). We measured $Ca^{2+}$ responses

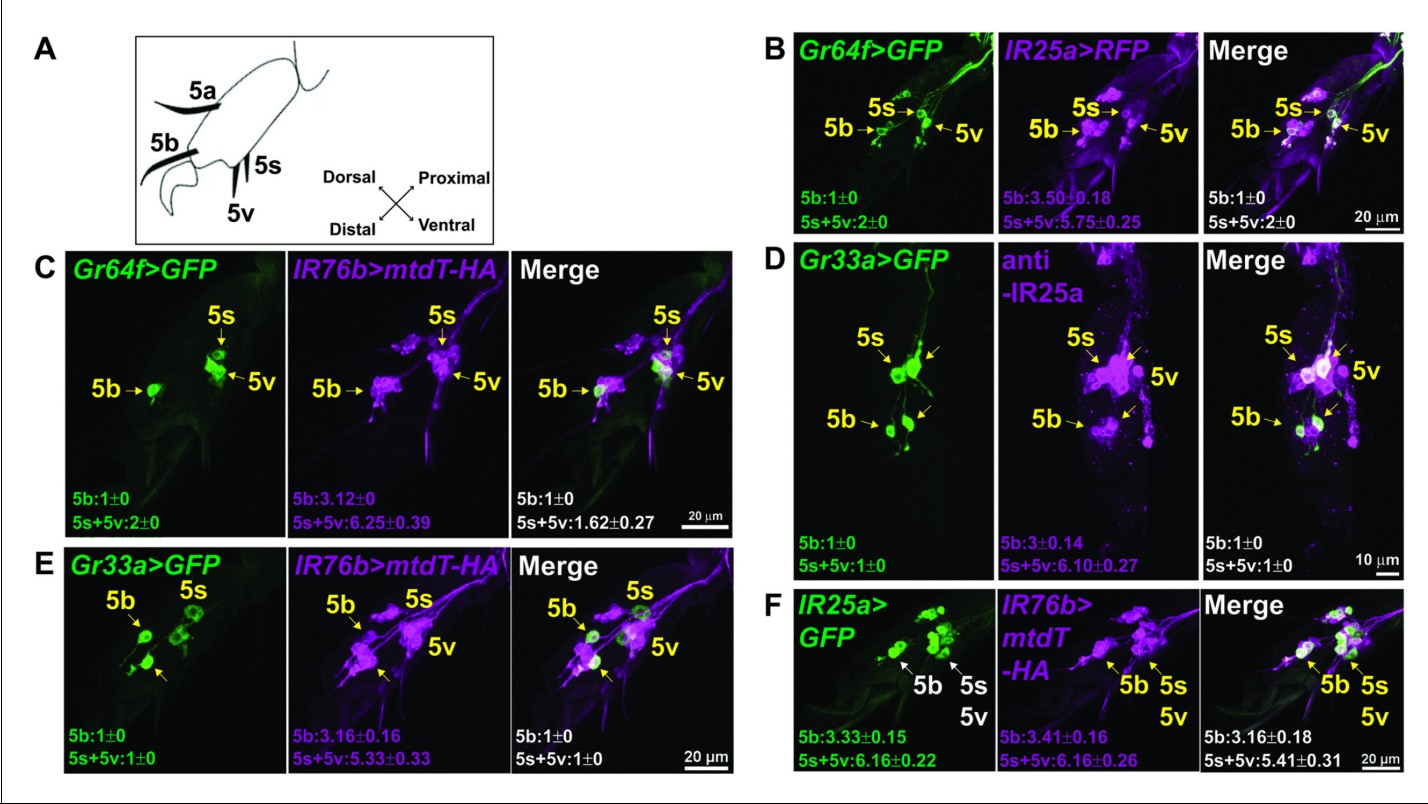

**Figure 2.** *IR25a* and *IR76b* are expressed in numerous taste neurons, including sweet and bitter GRNs. (**A**) Drawing of the fifth tarsal segment of the prothoracic leg that is used for immunofluorescence experiments (**B–F**). Tarsal taste sensilla are indicated. (**B and C**) *IR25a* and *IR76b* are expressed in sweet GRNs. Immunostaining with anti-GFP (green) and anti-mCD8 (B; magenta) or anti-HA (C; magenta) antibodies on whole-mount preparations of the fifth tarsal segment of the prothoracic leg from flies of the genotypes: *IR25a-GAL4/UAS-mCD8:RFP; Gr64f$^{LexA}$ lexAop-rCD2:GFP* (**B**) or *IR76b-QF UAS-mCD8:GFP/Gr64f-GAL4;QUAS-mtd-Tomato-3xHA/+* (**C**). Arrows refer to a GRN of a sensillum expressing both *Gr64f* and the indicated *IR* genes. (**D and E**) *IR25a* and *IR76b* are expressed in bitter GRNs. Immunostaining with anti-GFP (green) and anti-IR25a (D; magenta) or anti-HA (E; magenta) antibodies on whole-mount preparations of the fifth tarsal segment of the prothoracic leg from flies of the genotypes: *Gr33a$^{GAL4}$/UAS-mCD8:GFP* (**D**) or *IR76b-QF UAS-mCD8:GFP/Gr33a$^{GAL4}$;QUAS-mtd-Tomato-3xHA/+* (**E**). Arrows refer to a GRN of a sensillum expressing both *Gr33a* and the indicated *IR* genes. (**F**) *IR25a* and *IR76b* are largely co-expressed in tarsal GRNs. Immunostaining with anti-GFP (green) and anti-HA (magenta) antibodies on whole-mount preparations of the fifth tarsal segment of the prothoracic leg from flies of the genotypes: *IR76b-QF UAS-mCD8:GFP/IR25a-GAL4;QUAS-mtd-Tomato-3xHA/+*. Arrows refer to GRNs expressing both *IR25a* and *IR76b* genes. Numbers indicate the average count of IR or Gr expressing GRNs/sensillum. Due to close proximity, neurons could not always be associated with either the 5s and 5v sensillum, and the cell count was therefore pooled.
DOI: https://doi.org/10.7554/eLife.30115.008

The following figure supplement is available for figure 2:

**Figure supplement 1.** Co-expression of *IR25a* and *IR76b* with *Gr64f* and *Gr33a* in labellar GRNs.
DOI: https://doi.org/10.7554/eLife.30115.009

in sweet GRNs of the taste sensilla in the fifth tarsal segment of the prothoracic leg (5b, 5s and 5v; *Figure 4A*; the 5a sensillum is mainly dedicated to pheromone sensing, and no sugar or fatty acid responses were observed in these GRNs). In wild type flies, the sweet GRNs of 5b and 5s sensilla showed robust $Ca^{2+}$ increases to hexanoic and octanoic acids and modest, but significant increases to linoleic acid (*Figure 4B and C*). Only weak $Ca^{2+}$ responses were observed in the 5v-associated sweet GRNs. Importantly, $Ca^{2+}$ responses were absent in 5b- and 5s-associated sweet GRNs of *IR25a$^{-/-}$* or *IR76b$^{-/-}$* flies (*Figure 4D and E*). *IR25a* or *IR76b* requirement for cellular fatty acid responses was confirmed by transgene rescue experiments, which restored $Ca^{2+}$ increases to the level observed in *w$^{1118}$* controls flies (*Figure 4D and E*).

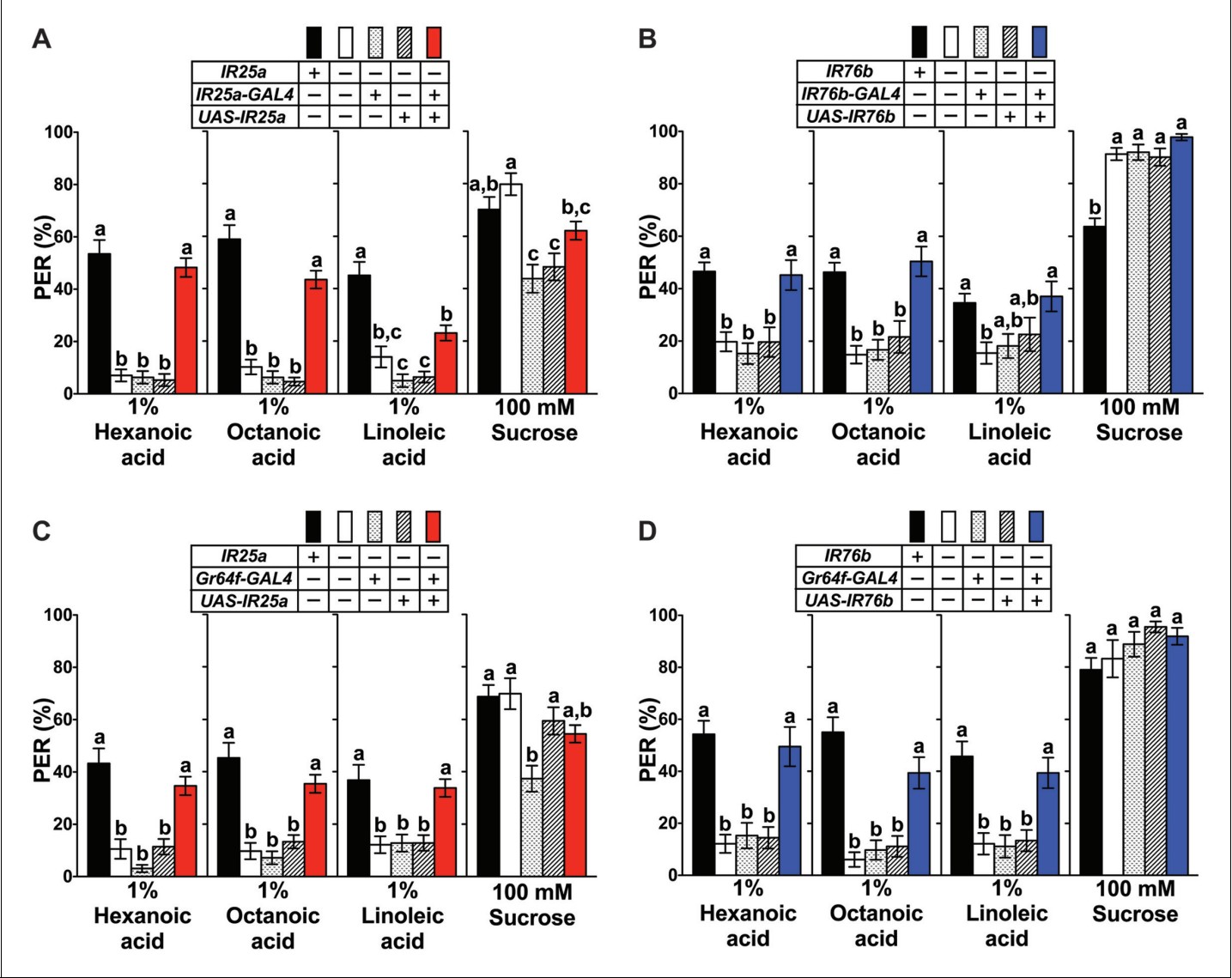

**Figure 3.** *IR25a* and *IR76b* are necessary for behavioral responses to fatty acids. (**A and B**) Mutations in *IR25a* or *IR76b* abolish PER responses to fatty acids. *IR25a* (**A**) and *IR76b* (**B**) mutant flies show significantly reduced PER responses to fatty acids, which are rescued by expression of *UAS-IR25a* and *UAS-IR76b* respectively, under control of their respective *GAL4* drivers. Expression of *UAS-IR25a* transgenes only partially rescues PER responses to linoleic acid (**A**). Each bar represents the mean ± SEM of PER responses (n = 34–116 flies). Bars with different letters are significantly different (Kruskal-Wallis test by ranks with Dunn's multiple comparison tests, p<0.05). Each y-axis delineates groups for Kruskal-Wallis test. Fly genotypes: wild-type: $w^{1118}$ (black), mutants: $IR25a^1/IR25a^1$ and $IR76b^1/IR76b^1$ (white), $IR25a^1$ IR25a-GAL4/IR25a$^1$ and IR76b-GAL4/+; IR76b$^1$/IR76b$^1$ (dotted), $IR25a^1$ UAS-IR25a/ $IR25a^1$ and UAS-IR76b/+; IR76b$^1$/IR76b$^1$ (lines), and rescue: $IR25a^1$ IR25a-GAL4/IR25a$^1$ UAS-IR25a (red), IR76b-GAL4/UAS-IR76b; IR76b$^1$/IR76b$^1$ (blue). (**C and D**) Functions of *IR25a* and *IR76b* in sweet GRNs are required and sufficient for fatty acid taste. *IR25a* (**C**) or *IR76b* (**D**) mutant flies show significantly reduced PER responses to fatty acids but not to sucrose. Restoring expression by *UAS-IR25a* (**C**) *or UAS-IR76b* (**D**) in sweet GRNs of *IR25a* or *IR76b* mutant flies is sufficient to rescue the loss of PER responses to fatty acids. Each bar represents the mean ± SEM of PER responses (n = 22–124 flies). Bars with different letters are significantly different (Kruskal-Wallis test by ranks with Dunn's multiple comparison tests, p<0.05). Each y-axis delineates groups for Kruskal-Wallis test. Fly genotypes: control flies: wild-type: $w^{1118}$ (black), mutants: $IR25a^1/IR25a^1$ and $IR76b^2/IR76b^2$ (white), $IR25a^1$ Gr64f-GAL4/IR25a$^1$ and Gr64f-GAL4/+; IR76b$^2$/IR76b$^2$ (dotted), $IR25a^1$ UAS-IR25a/IR25a$^1$ and UAS-IR76b/+; IR76b$^2$/IR76b$^2$ (lines), and rescue: $IR25a^1$ Gr64f-GAL4/IR25a$^1$ UAS-IR25a (red), Gr64f-GAL4/UAS-IR76b; IR76b$^2$/IR76b$^2$ (blue). Source data for summary graphs are provided in *Figure 3—source data 1*.
DOI: https://doi.org/10.7554/eLife.30115.010

The following source data is available for figure 3:

**Source data 1.** PER responses of *IR25a* and *IR76b* mutant flies to fatty acids.
DOI: https://doi.org/10.7554/eLife.30115.011

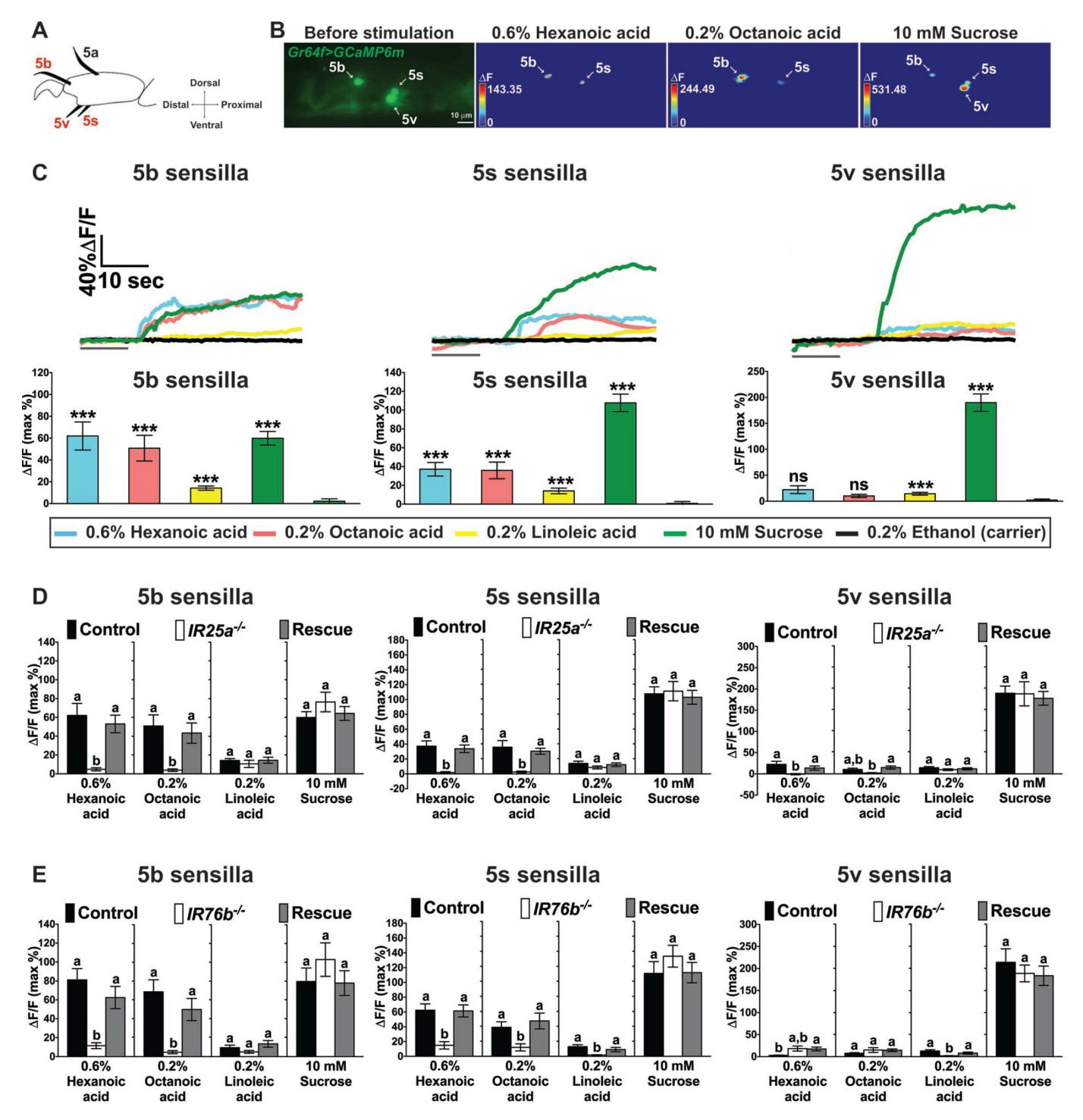

**Figure 4.** Cellular responses to fatty acids in sweet GRNs require *IR25a* and *IR76b* functions. (**A**) Diagram of the fifth tarsal segment of the prothoracic leg. Ca²⁺ imaging was carried out for the sweet GRN of the 5b, 5s and 5v sensilla (the *Gr64f-GAL4* expressing GRN in the 5a sensillum does not respond to sugar and was not included in our analysis). (**B**) Representative still images of the fifth tarsal segment of the prothoracic leg show maximum Ca²⁺ responses of the sweet GRNs upon stimulation by indicated ligands. ΔF indicates the changes in fluorescence light intensity of the cell body before/after ligand stimulation. (**C**) Representative fluorescence traces (top) and corresponding Ca²⁺ responses (bottom) of the sweet GRN associated with the 5b, 5s and 5v sensilla after stimulation with indicated ligands. The gray line underneath the fluorescence traces indicates time of ligand application. Hexanoic and octanoic acids elicit strong Ca²⁺ responses in the sweet GRN associated with 5b and 5s sensilla, but not the 5v sensillum. The 5v-associated sweet GRN shows weak, but significant Ca²⁺ responses to linoleic acid when compared to carrier. 10 mM sucrose was used as a

*Figure 4 continued on next page*

*Figure 4 continued*

positive control. Each bar represents the mean ± SEM of Ca$^{2+}$ imaging with 13–57 female prothoracic legs. Two-tailed, Mann-Whitney U test versus 0.2% ethanol (carrier), ***p<0.001, ns: not significant. Fly genotype is *Gr64f-GAL4/+; UAS-GCaMP6m/+*. Note that traces and graphs for a second *UAS-GCaMP6m* reporter (located on the second chromosome and used to analyze *IR76b* mutant flies) are shown in ***Figure 4—figure supplement 1A and B***. (D and E) *IR25a* and *IR76b* are necessary in sweet GRNs for fatty acid responses. Ca$^{2+}$ responses of 5b-, 5s- and 5v-associated sweet GRNs of *IR25a* (D) or *IR76b* (E) mutant flies to indicated ligands. Ca$^{2+}$ responses to hexanoic and octanoic acids are abolished in 5b- and 5s-associated sweet GRNs of both *IR25a* and *IR76b* mutant flies, and they are fully rescued when *UAS-IR25a* or *UAS-IR76b* expression is provided in sweet GRNs. Ca$^{2+}$ responses to linoleic acid are significantly reduced only in the sweet GRN associated with 5s and 5v sensilla from *IR76b* mutant flies (E). Each bar represents the mean ± SEM of Ca$^{2+}$ imaging with 16–50 female prothoracic legs. Bars with different letters are significantly different (Kruskal-Wallis test by ranks with Dunn's multiple comparison tests, p<0.05). Each y-axis delineates groups for Kruskal-Wallis test. Fly genotypes D: *Gr64f-GAL4/+; UAS-GCaMP6m/+* (Control, black bar), *IR25a$^2$ Gr64f-GAL4/IR25a$^2$; UAS-GCaMP6m/+* (*IR25a$^{-/-}$*, white) and *IR25a$^2$ Gr64f-GAL4/IR25a$^2$ UAS-IR25a; UAS-GCaMP6m/+* (Rescue, grey); Fly Genotypes E: *Gr64f-GAL4 UAS-GCaMP6m/+* (Control, black), *Gr64f-GAL4 UAS-GCaMP6m/+; IR76b$^2$/IR76b$^2$* (*IR76b$^{-/-}$*, white) and *Gr64f-GAL4 UAS-GCaMP6m/UAS-IR76b; IR76b$^2$/IR76b$^2$* (Rescue, grey). For representative traces of these genotypes, see ***Figure 4—figure supplement 1C and D***. Source data for summary graphs are provided in ***Figure 4—source data 1***.

DOI: https://doi.org/10.7554/eLife.30115.012

The following source data and figure supplements are available for figure 4:

**Source data 1.** Ca$^{2+}$ imaging results of sweet GRNs of *w$^{1118}$*, *IR25a* and IR76b mutant flies to fatty acids.

DOI: https://doi.org/10.7554/eLife.30115.015

**Figure supplement 1.** Ca$^{2+}$ responses of sweet GRNs to fatty acids.

DOI: https://doi.org/10.7554/eLife.30115.013

**Figure supplement 1—source data 1.** Ca$^{2+}$ imaging results of sweet GRNs.

DOI: https://doi.org/10.7554/eLife.30115.014

## The *IR56d* gene is sweet GRN specific and required for fatty acid taste

*IR25a* and *IR76b* are expressed in multiple GRNs per sensillum (***Figure 2***), including a recently characterized sour GRN where they are required to sense carboxylic acids and HCl (***Chen and Amrein, 2017***). This finding, together with the observation that IRs are likely to form tetrameric complexes containing three and possibly four different IR subunits (***Abuin et al., 2011***) led us to search for additional *IR* genes involved in fatty acid taste. We therefore examined seven candidate *IR* genes based on their expression in taste sensilla (***Cameron et al., 2010***; ***Koh et al., 2014***) for effects on PER when functionally perturbed. RNAi knock-down of one of these genes, *IR56d*, led to a significant reduction in PER responses to all fatty acids (***Figure 5A***), while knock-downs of or of mutations in the other six *IR* genes had no effect (***Figure 5—figure supplement 1***). We next expressed GCaMP6m in sweet GRNs, along with the *RNAi$^{IR56d}$* construct and carried out Ca$^{2+}$ imaging experiments of sweet GRNs. As expected, we observed an almost complete loss of GRN activation to hexanoic and octanoic acids (***Figure 5B and C***), similar as observed for *IR25a* or *IR76b* mutants (***Figure 4D and E***). Finally, we examined expression of an *IR56d-GAL4* line in the tarsal segments of the prothoracic leg (***Koh et al., 2014***). Indeed, antibody staining of tarsi of flies also expressing a marker for sweet GRNs showed complete overlap between *IR56d* and *Gr64f*, while no expression was observed in bitter GRNs, labeled by *Gr66a-LexA* (***Figure 5E and F***). Taken together, these experiments imply that the fatty acid taste receptor in sweet GRNs is composed of at least three subunits, IR25a, IR76b and the more narrowly expressed IR56d subunit. However, they do not exclude the possibility of yet a fourth IR being part of a fatty acid taste receptor complex.

## Bitter GRNs modulate acceptance behavior of hexanoic acid

Because *IR25a* and *IR76b* are also found in bitter GRNs, we carried out Ca$^{2+}$ imaging experiments using *Gr33a$^{GAL4}$*, a bitter GRN specific marker (***Moon et al., 2009***). Bitter GRNs of both the s- and b-type sensilla strongly responded to denatonium, a bitter tasting compound. Intriguingly, strong Ca$^{2+}$ increases were also observed in the 5b-associated bitter GRN to hexanoic acid, while octanoic or linoleic acid elicited no significant responses (***Figure 6A and B***; note that octanoic and linoleic acids could only be tested at 0.2%, due to low solubility; see Materials and methods). These observations indicate that at least hexanoic acid activates both appetitive and repulsive GRNs. However, and in contrast to sweet GRNs, bitter GRN responses to hexanoic acid did not require *IR25a* or *IR76b* (***Figure 6C and D***), indicating that these neurons employ a different molecular receptor.

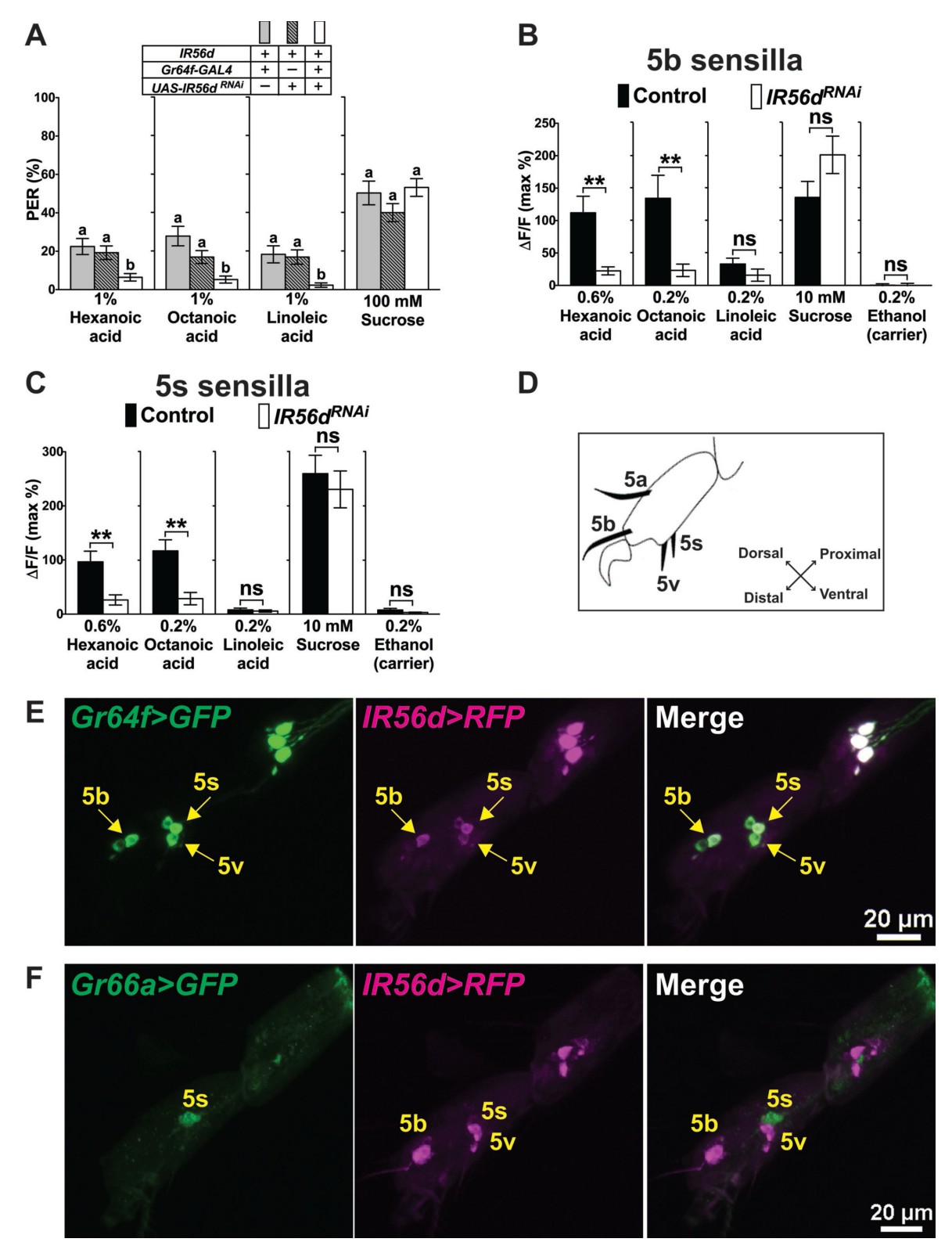

**Figure 5.** IR56d is necessary for fatty acid taste. (**A**) PER responses are reduced to all three fatty acids, but not to sucrose, in flies expressing an *UAS-RNAi^IR56d* construct in sweet GRNs. Targeted knockdown of *IR56d* is conducted by expression of *UAS-IR56d^RNAi* under control of *Gr64f-GAL4*. Each bar represents the mean ± SEM of PER responses (n = 49–59 flies). Bars with different letters are significantly different (Kruskal-Wallis test by ranks with Dunn's multiple comparison tests, p<0.05). Each y-axis delineates groups for Kruskal-Wallis test. Fly genotypes: control flies: *Gr64f-GAL4/+* (grey), *UAS-*

*Figure 5 continued on next page*

*Figure 5 continued*

*IR56d$^{RNAi}$/+* (lines) and *IR56d* knock-down fly: *Gr64f-GAL4/+; UAS-IR56d$^{RNAi}$/+* (white). (**B and C**) Ca$^{2+}$ responses in 5b- (**B**) and 5s- (**C**) associated sweet GRNs of flies expressing *UAS-GCaMP6m* and *UAS-RNAi$^{IR56d}$* in sweet GRNs show loss of fatty acid induced neural activation, compared to neurons of control flies. Targeted knockdown of *IR56d* in sweet GRNs has no effect on Ca$^{2+}$ responses to linoleic acid. 10 mM sucrose was used as a positive control. Each bar represents the mean ± SEM of Ca$^{2+}$ imaging with 3–21 female prothoracic legs. Asterisks indicate a significant difference between the *IR56d$^{RNAi}$* and control flies (Two-tailed, Mann-Whitney U test, \*\*p<0.01, ns: not significant). Each y-axis delineates groups for Mann-Whitney U test. Fly genotypes: *Gr64f-GAL4 UAS-GCaMP6m/+* (control, black) and *Gr64f-GAL4 UAS-GCaMP6m/+; UAS-IR56d$^{RNAi}$/+* (*IR56d$^{RNAi}$*, white). (**D**) Drawing of the fifth tarsal segment of the prothoracic leg that is used for immunofluorescence experiments (**E and F**). Tarsal taste sensilla are indicated. (**E and F**) Expression of *IR56d-GAL4* is restricted to a single GRN in all sensilla examined and co-localizes with *Gr64f-LexA* (**E**). No expression of *IR56d-GAL4* is observed in *Gr66a-LexA* expressing bitter GRNs associated with 5s sensilla (**F**). Immunostaining with anti-GFP (green) and anti-mCD8 (magenta) antibodies on whole-mount preparations of tarsal segments of the prothoracic leg from flies of the genotypes: *UAS-mCD8:RFP lexAop-rCD2:GFP; IR56d-GAL4/+;Gr64f$^{LexA}$/TM6c* (**E**) and *UAS-mCD8:RFP lexAop-rCD2:GFP;Gr66a-LexA/IR56d-GAL4;+/TM6c* (**F**). Arrows refer to a GRN of a sensillum expressing both *Gr64f* and *IR56d* genes. Source data for summary graphs are provided in *Figure 5—source data 1*.

DOI: https://doi.org/10.7554/eLife.30115.016

The following source data and figure supplements are available for figure 5:

**Source data 1.** PER and Ca$^{2+}$ responses of *IR56d* knock-down flies to fatty acids.

DOI: https://doi.org/10.7554/eLife.30115.019

**Figure supplement 1.** PER responses in *IR* knockdown flies (**A**) or *IR* mutant flies (**B**) to fatty acids.

DOI: https://doi.org/10.7554/eLife.30115.017

**Figure supplement 1—source data 1.** PER responses of flies with knockdown of *IR* genes in sweet GRNs (**A**) or *IR* mutants (**B**) to fatty acids.

DOI: https://doi.org/10.7554/eLife.30115.018

To investigate hexanoic acid response characteristics in more detail, we compared dose response profiles of sweet and bitter GRNs of 5b sensilla (*Figure 7A*). Interestingly, sweet GRNs reached maximal activation already at the modest concentration of 1% (*Figure 7A*, left). In contrast, maximal activation of bitter GRNs was reached at the highest applicable concentration (2.5%; *Figure 7A,* right). These observations suggest that bitter GRNs might counteract the activity of sweet GRNs above a certain concentration threshold (>1%) and possibly modulate taste behavior. Indeed, *w$^{1118}$* flies exhibited a maximal PER response at ~1%, whereas higher concentrations of hexanoic acid led to a decrease in PER responses (*Figure 7B*). In contrast, when we inhibited the activity of bitter GRNs by either expressing TNT or the inward-rectifier potassium channel Kir2.1, PER responses stayed at the same high level or continued to increase with increasing hexanoic acid concentration (*Figure 7C* and *Figure 7—figure supplement 2*). Thus, activation of bitter GRNs appears to suppress PER responses at high (2.5%) hexanoic acid concentration. To test whether the decrease of sweet GRN responses was mediated through lateral inhibition, we compared Ca$^{2+}$ responses of sweet GRNs to 2.5% hexanoic acid in the presence and absence of functional bitter GRNs. However, hexanoic acid elicited the same Ca$^{2+}$ responses in sweet GRNs, regardless of whether functional bitter GRNs were present (*Figure 7D*).

## Discussion

*IR* genes have emerged as a second large gene family encoding chemoreceptors in insects. In the *Drosophila* olfactory system, IRs function as multimeric receptors in coeloconic olfactory sensory neurons (OSN) and are thought to sense volatile carboxylic acids, amines and aldehydes (*Abuin et al., 2011*; *Benton et al., 2009*; *Hussain et al., 2016*; *Silbering et al., 2011*). Expression analyses have shown that each coeloconic OSN expresses up to four IR genes, including high levels of either IR8a or IR25a (*Abuin et al., 2011*). IR25a and IR8a are distinct from other IRs in that they are more conserved to each other and iGluRs, and they share a long amino terminal domain absent in all other IRs (*Croset et al., 2010*). These observations, along with functional analyses of basiconic olfactory neurons that express combinations of *IR* genes, led to a model in which IR based olfactory receptors are tetrameric complexes thought to consist of up to three different subunits that contain at least one core unit (IR8a or IR25a) and two additional IRs that determine ligand binding specificity (*Abuin et al., 2011*; *Silbering et al., 2011*). The findings presented in this paper expand this concept to taste receptors that sense fatty acids through the sweet GRNs found in tarsal taste sensilla.

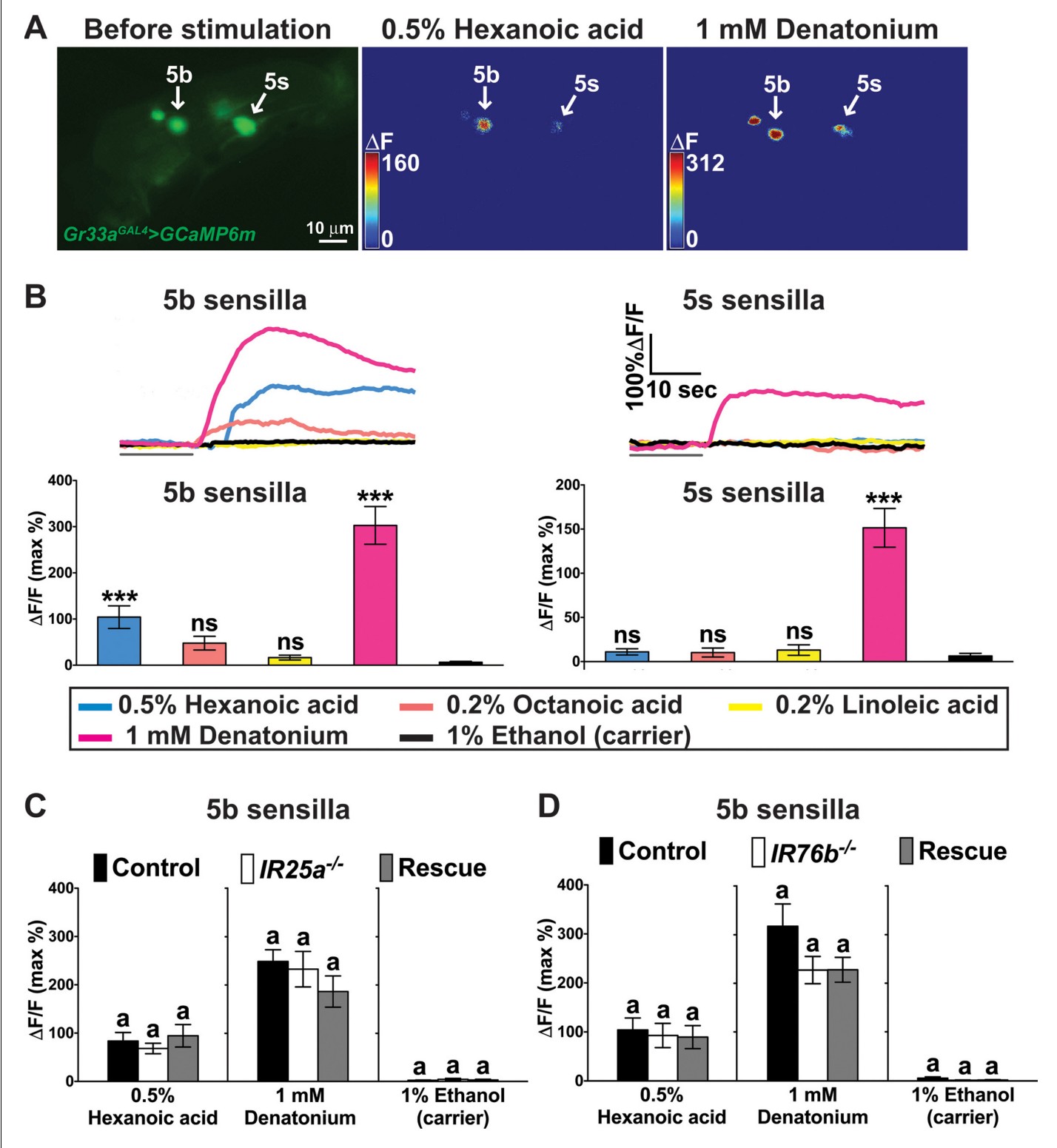

**Figure 6.** A subset of bitter GRNs responds to hexanoic acids in an *IR25a/IR76b* independent manner. (**A**) Representative still images of the fifth tarsal segment of the prothoracic leg show maximum $Ca^{2+}$ responses in the bitter GRNs associated with 5b and 5 s sensilla upon stimulation by indicated ligands. ΔF indicates the changes in fluorescence light intensity of the cell body before/after ligand application. (**B**) Representative fluorescence traces (top) and corresponding $Ca^{2+}$ responses (bottom) of the 5b- and 5s-associated bitter GRNs upon stimulation by indicated ligands. The gray line underneath the fluorescence traces indicates time of ligand application. Hexanoic acid elicits highly significant $Ca^{2+}$ responses in the 5b- associated

*Figure 6 continued on next page*

Figure 6 continued

bitter GRNs (compared to carrier). 1 mM denatonium was used as a positive ligand control. 1% ethanol was used as a carrier to facilitate solubilization of high concentrations of hexanoic acid (2.5%). Each bar represents the mean ± SEM of $Ca^{2+}$ imaging with 10–30 female prothoracic legs. Two-tailed, Mann-Whitney U test versus carrier (1% ethanol), ***$p<0.001$, ns: not significant. Fly genotype is $Gr33a^{GAL4}$ UAS-GCaMP6m/+. (C and D) Cellular responses to fatty acid in bitter GRNs do not require $IR25a$ and $IR76b$. $Ca^{2+}$ responses of the 5b-assoicated bitter GRNs of $IR25a$ (C) or $IR76b$ (D) mutant flies to indicated ligands. $Ca^{2+}$ responses of GRNs of mutants to hexanoic acid were not significantly reduced when compared to control flies. Each bar represents the mean ± SEM of $Ca^{2+}$ imaging with 4–42 female prothoracic legs. Bars with different letters are significantly different (Kruskal-Wallis test by ranks with Dunn's multiple comparison tests, $p<0.05$). Each y-axis delineates groups for Kruskal-Wallis test. Fly genotypes C: $Gr33a^{GAL4}$/+; UAS-GCaMP6m/+ (Control, black), $IR25a^2$ $Gr33a^{GAL4}$/$IR25a^2$; UAS-GCaMP6m/+ ($IR25a^{-/-}$, white), and $IR25a^2$ $Gr33a^{GAL4}$/$IR25a^2$ UAS-IR25a; UAS-GCaMP6m/+ (Rescue, grey); Fly Genotypes D: $Gr33a^{GAL4}$ UAS-GCaMP6m/+ (Control, black), $Gr33a^{GAL4}$ UAS-GCaMP6m/+; $IR76b^2$/$IR76b^2$ ($Ir76b^{-/-}$, white) and $Gr33a^{GAL4}$ UAS-GCaMP6m/UAS-IR76b; $IR76b^2$/$IR76b^2$ (Rescue, grey). See **Figure 6—figure supplement 1—source data 1** for hexanoic acid responses of 5s-associated bitter GRNs from $IR25a$ or $IR76b$ mutant flies. Source data for summary graphs are provided in **Figure 6—source data 1**.

DOI: https://doi.org/10.7554/eLife.30115.020

The following source data and figure supplements are available for figure 6:

**Source data 1.** $Ca^{2+}$ responses of bitter GRNs of $IR25a$ or $IR76b$ mutant flies to fatty acids.
DOI: https://doi.org/10.7554/eLife.30115.023
**Figure supplement 1.** Hexanoic acid responses of 5s -associated bitter GRNs of $IR25a$ (A) or $IR76b$ (B) mutant flies.
DOI: https://doi.org/10.7554/eLife.30115.021
**Figure supplement 1—source data 1.** $Ca^{2+}$ responses of bitter GRNs associated with 5s sensilla of $IR25a$ (A) or $IR76b$ (B) mutant flies to hexanoic acid.
DOI: https://doi.org/10.7554/eLife.30115.022

## IR25a and IR76b mediate fatty acid taste in sweet GRNs

Our analysis extends the multimodal role of IR25a and IR76b to the taste systems. Consistent with gene expression arrays (*Cameron et al., 2010*), we showed in this paper that up to three GRNs, including many sweet and bitter GRNs, co-express $IR25a$ and $IR76b$. Our functional studies have established a novel role for these two IR proteins in fatty acid taste, which revealed that these two subunits are not only critically important to elicit PER responses in flies when challenged with fatty acids, but are also necessary for fatty acid induced $Ca^{2+}$ increases in tarsal sweet GRNs. Based on these findings and with consideration of their established role in other sensory systems, we propose that IR25a and IR76b play central roles in sweet GRNs in a multimeric receptor complex for initiating appetitive taste behavior to these chemicals. Intriguingly, both genes are also co-expressed in two other GRNs of most tarsal taste sensilla, strongly arguing for additional taste functions. While the subset of tarsal bitter GRNs activated by hexanoic acid does not require either gene, we have discovered that the third GRN (the sour GRN) is narrowly tuned to acids in an $IR25a/IR76b$ dependent manner (*Chen and Amrein, 2017*). These observations suggest that modality specific IRs are likely expressed in a cell-type specific fashion whereby they complement IR25a/IR76b to function as either a fatty acid or a sour taste receptor. Indeed, our screen identified $IR56d$, a gene that is expressed in sweet GRNs of tarsal taste sensilla, as a likely candidate encoding an IR subunit specific for a fatty acid taste receptor (**Figure 5**). It remains to be seen whether IR25a, IR76b and IR56d comprise all subunits that constitute this receptor or whether yet additional IRs are necessary to mediate responses to these chemicals.

The fact that different food chemicals can activate a single class of neurons raises the question how flies discriminate between sugars and fatty acids. First, we note the difference in sensitivity of appetitive GRNs to sugars and fatty acids, respectively: The most responsive GRN for sugars is the one associated with the 5v sensilla, followed by that with the 5s and finally the 5b sensilla, while the responsiveness for fatty acids is the reverse (5b > 5s > 5v; **Figure 4**). Second, fatty acids induces weaker PER responses from stimulation of the labial palps as opposed to tarsi, while sugars induce equally strong PER responses from stimulation of either taste organ (**Figure 1—figure supplement 1**) (*Fujii et al., 2015*). Third, at least some fatty acids activate bitter GRNs (**Figure 6**), and hence, generate more complex activation patterns in the brain than sugars, which are not known to activate neurons other than sweet GRNs. These properties may provide a rationale for differential coding of these two classes of chemicals in the brain. Finally, sugars but not fatty acids are soluble in water, and hence, the specific solvents in which these chemicals are presented provides different textural quality, which was recently shown to play a role in taste perception (*Zhang et al., 2016*).

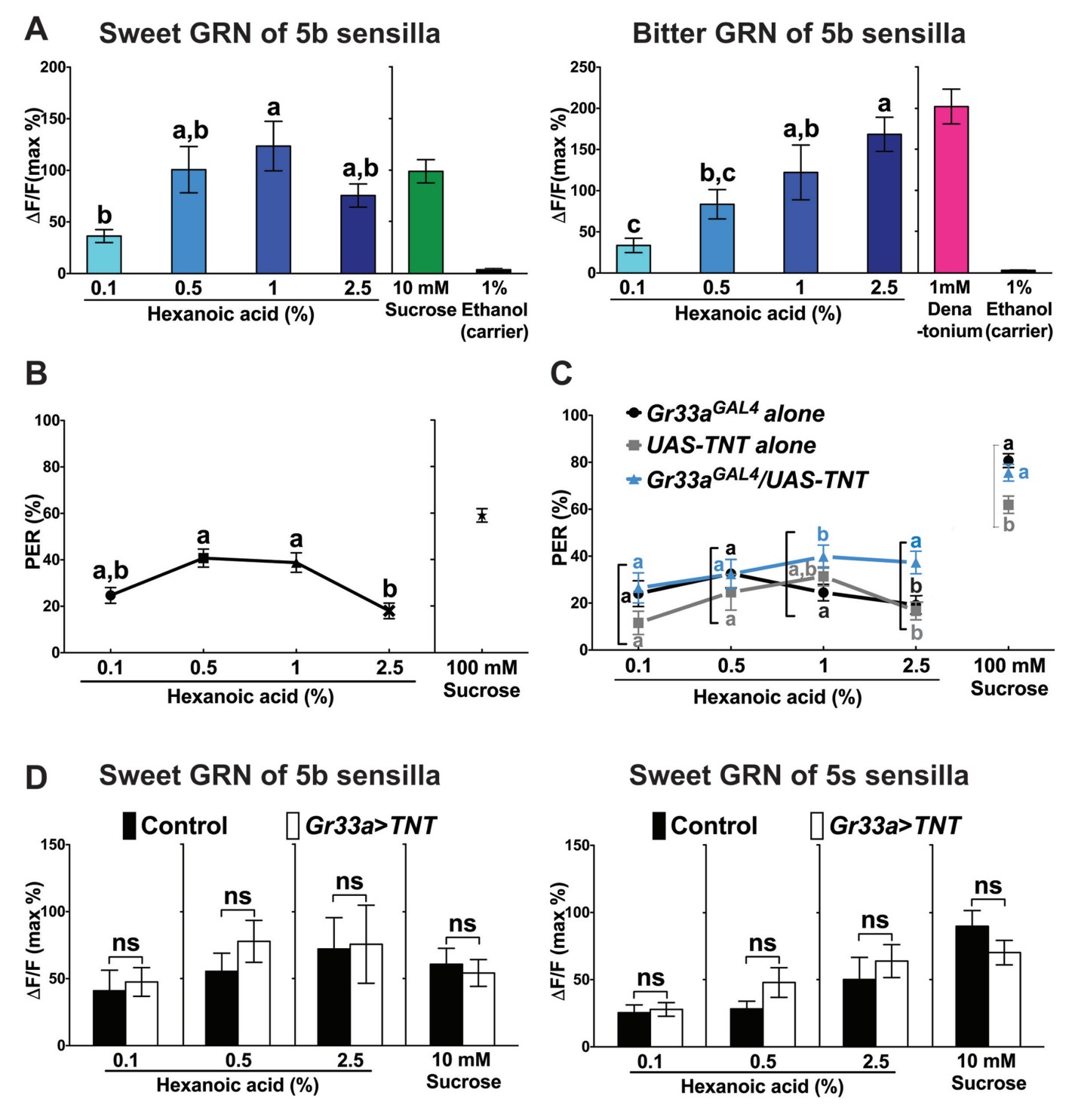

**Figure 7.** Bitter GRNs inhibit acceptance of high concentration of hexanoic acid. (**A**) Hexanoic acid dose response profiles of the sweet (left) and the bitter (right) GRNs associated with the 5b sensilla. sweet GRN reaches a maximal response already at 1%, while bitter GRNs responses further increases as ligand concentration increases. Each bar represents the mean ± SEM of Ca$^{2+}$ imaging with 7–30 female prothoracic legs. Bars with different letters are significantly different (Kruskal-Wallis test by ranks with Dunn's multiple comparison tests, p<0.05). Each y-axis delineates groups for Kruskal-Wallis test. Fly genotypes: *Gr64f-GAL4 UAS-GCaMP6m/+* and *Gr33a$^{GAL4}$/+;UAS-GCaMP6m/+*. For hexanoic acid response profiles of the sweet GRNs, or the bitter GRN associated with 5 s/5v sensilla, see *Figure 7—figure supplement 1*. (**B**) PER responses of wild-type flies (*w$^{1118}$*) to different concentrations of hexanoic acid. At high concentration (2.5%) hexanoic acids induces a much lower PER response compared to more modest concentrations (0.5%–1%). Each symbol represents the mean ± SEM of PER responses (n = 61–142 flies). Symbols with different letters are significantly different (Kruskal-Wallis test

*Figure 7 continued on next page*

*Figure 7 continued*

by ranks with Dunn's multiple comparison tests, p<0.01). Each y-axis delineates groups for Kruskal-Wallis test. (**C**) Inactivation of bitter GRN leads to concentration dependent PER responses to hexanoic acid, while control flies show a response profile similar to $w^{1118}$ flies (see B). Each symbol represents the mean ± SEM of PER responses (n = 23–121 flies). Symbols with different letters are significantly different (Kruskal-Wallis test by ranks with Dunn's multiple comparison tests, p<0.05). Each square bracket delineates groups for Kruskal-Wallis test. Fly genotypes: $Gr33a^{GAL4}$/+ ($Gr33a^{GAL4}$ alone), *UAS-TNT/+* (*UAS-TNT* alone) and $Gr33a^{GAL4}$/*UAS-TNT*. (**D**) Inactivation of bitter GRNs has no effect on cellular responses of sweet GRNs to hexanoic acid. $Ca^{2+}$ responses of sweet GRNs to 2.5% hexanoic acid are similar regardless of whether a functional bitter GRN is present. Bitter GRNs was inactivated by expressing *UAS-TNT* under the control of $Gr33a^{GAL4}$. Each bar represents the mean ± SEM of 10–15 female GRNs from prothoracic legs. Two-tailed, Mann-Whitney U test between the flies lacking functional bitter GRNs and control flies, p<0.05, ns: not significant. Each y-axis delineates groups for Mann-Whitney U test. Fly genotypes: $Gr64f^{LexA}$/*lexAop-GCaMP6m* (control) and $Gr33a^{GAL4}$/*UAS-TNT; Gr64f^{LexA}/lexAop-GCaMP6m* (*Gr33a > TNT*). Source data for summary graphs are provided in *Figure 7—source data 1*.

DOI: https://doi.org/10.7554/eLife.30115.024

The following source data and figure supplements are available for figure 7:

**Source data 1.** Dosage dependent $Ca^{2+}$ and PER responses of neurons and flies to hexanoic acid.

DOI: https://doi.org/10.7554/eLife.30115.029

**Figure supplement 1.** Hexanoic acid responses of 5s/5v associated sweet (**A and B**) or bitter (**C**) GRNs.

DOI: https://doi.org/10.7554/eLife.30115.025

**Figure supplement 1—source data 1.** $Ca^{2+}$ responses of sweetGRNs associated with 5s (**A**) or 5v (**B**) sensilla to different dosages of hexanoic acid.

DOI: https://doi.org/10.7554/eLife.30115.026

**Figure supplement 2.** Bitter GRNs suppress PER responses to high concentration of hexanoic acid.

DOI: https://doi.org/10.7554/eLife.30115.027

**Figure supplement 2—source data 1.** Dosage dependent PER responses of flies with inactivated bitter GRNs to hexanoic acid.

DOI: https://doi.org/10.7554/eLife.30115.028

*NorpA*, which encodes a phospholipase C (PLC), plays a critical role in sweet GRNs for appetitive feeding responses to fatty acids, but it is dispensable for behavioral responses to sugars (*Masek and Keene, 2013*). We found that its absence also selectively abolishes $Ca^{2+}$ responses to fatty acids, but not sugars, in sweet cells (*Figure 1—figure supplement 2*). *NorpA* is known for its role as downstream effector of G-protein coupled receptors in the fly's visual system (*Bloomquist et al., 1988*; *Wilson and Ostroy, 1987*), but interestingly it is also required for olfactory responses in neurons of the maxillary palps (*Riesgo-Escovar et al., 1995*), which express ORs that are thought to function as ligand-gated ion channels (*Sato et al., 2008*; *Wicher et al., 2008*). We note that fatty acid taste in mammals is in part mediated by two G – protein coupled receptors, GPR40 and GPR120 (*Cartoni et al., 2010*; *Matsumura et al., 2007*), and that one of these (GPR120) was found to signal through a phospholipase C (*Reed and Xia, 2015*). Thus, future studies will be necessary to gain insights for how PLC mediates chemosensory responses through ORs and the phylogenetically unrelated IRs.

## IRs form multimodal receptors that mediate diverse sensory cues

Mutlimeric IR based receptors were recently shown to be required in non-chemosensory processes. Specifically, Dorsal Organ Cool Cells (DOCCs) located in the larval brain, express and require the function of three IRs (IR21a IR25a and IR93a), thereby allowing larvae to avoid temperatures below ~20°C (*Knecht et al., 2016*; *Ni et al., 2016*). Similarly, two sets of cells in the antennal sacculus of adult flies, requiring the functions of IR25a and IR93a and either IR40a or IR68a, were shown to mediate a fly's preferred humidity environment, which is generally in the dry range, but is also dependent on the fly's hydration state (*Enjin et al., 2016*; *Knecht et al., 2017*; *Knecht et al., 2016*). Intriguingly, these non-chemosensory IR complexes share a common theme with the fatty acid and carboxylic acid taste receptors (*Chen and Amrein, 2017*) in that they all require a core unit (IR25a) and two additional IRs that mediate specificity for a particular stimulus type (i.e. temperature, humidity, fat, acid).

An IR76b based sodium channel and an IR76b based amino acid receptor appear to lack an obligate core unit (IR25a or IR8a) found in olfactory receptors or fatty acid and carboxylic acid taste receptors. The IR76b sodium channel mediates salt responses in a heterologous systems independently of any other IRs (*Zhang et al., 2013*), while a proposed multimeric IR76b containing receptor mediates amino acids taste in wild type and *IR25a* mutant flies (*IR8a* is not expressed in taste

neurons) (*Ganguly et al., 2017*). It will be interesting to elucidate the compositions of complete IR based amino acid and sour taste receptors, and – with regard of amino acid receptors – to identify the neurons that mediate this taste modality.

# Materials and methods

**Key resources table**

| Reagent type | Designation | Source or reference | Identifiers |
|---|---|---|---|
| Antibodies | | | |
| Chicken polyclonal anti-GFP | Anti-GFP | Thermo Fisher Scientific | PA1-86341; RRID: AB_931091 |
| Rat monoclonal anti-mCD8 | Anti-mCD8 | Thermo Fisher Scientific | MCD0800; RRID: AB_10392843 |
| Rabbit polyclonal anti-IR25a | Anti-IR25a | *Benton et al. (2009)* | N/A |
| Mouse monoclonal anti-HA | Anti-HA | Covance Research Product Inc. | MMS-101P; RRID: AB_2314672 |
| Alexa 488 conjugated goat anti-chicken | Green | Thermo Fisher Scientific | A11039; RRID: AB_2534096 |
| Cy3-conjugated goat anti-rat | Magenta | Jackson Immunoresearch Laboratories Inc. | 112-165-072; RRID: AB_2338248 |
| Cy3-conjugated goat anti-rabbit | Magenta | Jackson Immunoresearch Laboratories Inc. | 111-166-003; RRID: AB_2338000 |
| Cy3-conjugated goat anti-mouse | Magenta | Jackson Immunoresearch Laboratories Inc. | 115-166-072; RRID: AB_2338706 |
| Chemical compounds | | | |
| Hexanoic acid | Hexanoic acid | Sigma-Aldrich | H12137 |
| Octanoic acid | Octanoic acid | Sigma-Aldrich | O3907 |
| Linoleic acid | Linoleic acid | Sigma-Aldrich | L1376 |
| Denatonium benzoate | Denatonium | Sigma-Aldrich | D5765 |
| Sucrose | Sucrose | Amresco | M1117 |
| Experimental Models: Organisms/Strains | | | |
| *D. melanogaster: w1118* | Wild-type control | Bloomington *Drosophila* Stock Center | BDSC: 3605; FlyBase: FBst0003605 |
| *D. melanogaster: w\*; P{Gr64f-GAL4.9.7}5/CyO; MKRS/TM2* | Gr64f-GAL4 | *Dahanukar et al. (2007)* | Flybase: FBst0057669 |
| *D. melanogaster: w\*; TI{LexA::VP16}Gr64f^LexA* | Gr64f^LexA | *Fujii et al. (2015)* | Flybase: FBti0168176 |
| *D. melanogaster: w\*; TI{GAL4}Gr33a^GAL4* | Gr33a^GAL4 | *Moon et al. (2009)* | Flybase: FBst0031425 |
| *D. melanogaster: w\*; P{UAS-TeTxLC.tnt}G2* | UAS-TNT | *Sweeney et al. (1995)* | Flybase: FBst0028838 |
| *D. melanogaster: w\*; P{lexAop-rCD2-GFP}* | lexAop-rCD2:GFP | *Lai and Lee (2006)* | Flybase: FBst0066687 |
| *D. melanogaster: w\*; P{10XUAS-IVS-mCD8::RFP}attP40* | UAS-mCD8:RFP | Bloomington *Drosophila* Stock Center | BDSC: 32219; Flybase: FBti0131967 |
| *D. melanogaster: w1118; PBac{20XUAS-IVS-GCaMP6m}VK00005* | UAS-GCaMP6m | Bloomington *Drosophila* Stock Center | BDSC: 42750; Flybase: FBst0042750 |
| *D. melanogaster: y1 w\*; P{UAS-mCD8::GFP.L}LL5, P{UAS-mCD8::GFP.L}2* | UAS-mCD8:GFP | Bloomington *Drosophila* Stock Center | BDSC: 5137; Flybase: FBst0005137 |
| *D. melanogaster: y1 w1118; P{QUAS-mtdTomato-3xHA}26* | QUAS-mtd-Tomato-3xHA | Bloomington *Drosophila* Stock Center | BDSC: 30005; Flybase: FBst0030005 |

*Continued on next page*

Continued

| Reagent type | Designation | Source or reference | Identifiers |
|---|---|---|---|
| D. melanogaster: $y^1$ sc* $v^1$; P{TRiP.HMC03664}attP40 | UAS-IR94h-RNAi | Bloomington Drosophila Stock Center | BDSC: 53675; Flybase: FBst0053675 |
| D. melanogaster: $w^{1118}$; Mi{ET1}IR52c$^{MB04402}$ | Mi{ET1}IR52c$^{MB04402}$ | Bloomington Drosophila Stock Center | BDSC: 24580; Flybase: FBst0024580 |
| D. melanogaster: $w^{1118}$; Mi{ET1}IR56b$^{MB09950}$ | Mi{ET1}IR56b$^{MB09950}$ | Bloomington Drosophila Stock Center | BDSC: 27818; Flybase: FBst0027818 |
| D. melanogaster: $y^1$ w*; Mi{MIC}IR62a$^{MI00895}$ lml1$^{MI00895}$/TM3, Sb$^1$, Ser$^1$ | Mi{Mic}IR62a$^{MI00895}$ | Bloomington Drosophila Stock Center | BDSC: 32713; Flybase: FBst0032713 |
| D. melanogaster: $w^{1118}$; P{UAS-norpA.WT}2 | UAS-norpA | Bloomington Drosophila Stock Center | BDSC: 35529; Flybase: FBst0035529 |
| D. melanogaster: w*; TI{TI}IR25a$^1$/CyO | IR25a$^1$/CyO | Benton et al. (2009) | Flybase: FBst0041737 |
| D. melanogaster: w*; P{IR25a-GAL4.A}236.1; TM2/TM6B, Tb$^1$ | IR25a-GAL4 | Abuin et al. (2011) | Flybase: FBst0041728 |
| D. melanogaster: w*; M{UAS-IR25a.attB} | UAS-IR25a | Abuin et al. (2011) | Flybase: FBal0249355 |
| D. melanogaster: {KK104276}VIE-260B | UAS-IR11a-RNAi | Vienna Drosophila Resource Center | VDRC ID: 100422; Flybase: FBgn0030385 |
| D. melanogaster: $w^{1118}$; P{GD773}v2472 | UAS-IR21a-RNAi | Vienna Drosophila Resource Center | VDRC ID: 2472; Flybase: FBgn0031209 |
| D. melanogaster: $w^{1118}$; P{GD2094}v4704 | UAS-IR56b-RNAi | Vienna Drosophila Resource Center | VDRC ID: 4704; Flybase: FBgn0034456 |
| D. melanogaster: $w^{1118}$; P{GD2096}v6112 | UAS-IR56d-RNAi | Vienna Drosophila Resource Center | VDRC ID: 6112; Flybase: FBgn0034458 |
| D. melanogaster: w*; IR76b$^1$ | IR76b$^1$/IR76b$^1$ | Zhang et al. (2013) | Flybase: FBst0051309 |
| D. melanogaster: w*; IR76b$^2$ | IR76b$^2$/IR76b$^2$ | Zhang et al. (2013) | Flybase: FBst0051310 |
| D. melanogaster: w*; P{IR76b-GAL4.1.5}2 | IR76b-GAL4, | Zhang et al. (2013) | Flybase: FBst0051311 |
| D. melanogaster: w*; P{UAS-IR76b.Z}2/CyO; TM2/TM6B, Tb$^1$ | UAS-IR76b | Zhang et al. (2013) | Flybase: FBst0052610 |
| D. melanogaster: w*; P{IR76b-QF.1.5} | IR76b-QF | Zhang et al. (2013) | Flybase: FBtp0085487 |
| D. melanogaster: w*; P{UAS-Hsap\KCNJ2.EGFP}1 | UAS-Kir2.1-GFP | Baines et al. (2001); Paradis et al. (2001) | Flybase: FBst0006596 |
| D. melanogaster: w*; norpA$^{36(P24)}$ | norpA$^{P24}$ | Masek and Keene (2013) | Flybase: FBst0009048 |
| D. melanogaster: w*; P{IR56d-GAL4.K}7–2/CyO; P{UAS-mCD8::GFP.L}LL6 | IR56d-GAL4 | Koh et al. (2014) | Flybase: FBst0060708 |
| D. melanogaster: w*; Gr66a-LexA/CyO; TM2/TM6B | Gr66a-LexA | Thistle et al. (2012) | Flybase: FBal0277069 |
| Software and Algorithms | | | |
| NIS-Elements | N/A | Nikon | N/A |
| Prism software | Prism software | GraphPad Software 5.0 Inc | N/A |
| Other | | | |
| Nikon Eclipse Ti inverted microscope | Nikon Eclipse Ti inverted microscope | Nikon | N/A |
| Nikon A1 confocal microscope | Nikon A1R confocal microscope system | Nikon | N/A |
| 35 mM Glass bottom dish | Glass bottom culture dish | MatTek Corporation | P35G-0–10 C |

### Drosophila stocks

Flies were maintained on standard corn meal food in plastic vials under a 12 hr light/dark cycle at 25° C. The $w^{1118}$ strain was used as a wild-type control. Fly lines used: *Gr64f-Gal4* (*Dahanukar et al., 2007*); *Gr64f^LexA* (*Fujii et al., 2015*); *Gr33a^GAL4* (*Moon et al., 2009*); *UAS-TNT* (*Sweeney et al., 1995*); *lexAop-rCD2:GFP* (*Lai and Lee, 2006*); *UAS-mCD8:RFP, UAS-GCaMP6m, UAS-mCD8:GFP, QUAS-mtd-Tomato-3xHA, UAS-IR94h-RNAi, Mi{ET1}IR52c^MB04402, Mi{ET1}IR56b^MB09950, Mi{Mic} IR62a^MI00895* and *UAS-norpA* (Bloomington Drosophila Stock Center, numbers 32219, 42750, 5137, 30005, 53675, 24580, 27818, 32713 and 35529); *IR25a^1* (*Benton et al., 2009*); *IR25a-GAL4, UAS-IR25a* (*Abuin et al., 2011*); *UAS-IR11a-RNAi, UAS-IR21a-RNAi, UAS-IR56b-RNAi* and *UAS-IR56d-RNAi* (Vienna Drosophila Resource Center, transformant ID 100422, 2472, 4704 and 6112); *IR76b^1, IR76b^2, IR76b-GAL4, UAS-IR76b, IR76b-QF* (*Zhang et al., 2013*); *UAS-Kir2.1-GFP* (*Baines et al., 2001; Paradis et al., 2001*); *norpA^P24* (*Masek and Keene, 2013*); *IR56d-GAL4* (*Koh et al., 2014*) and *Gr66a-LexA* (*Thistle et al., 2012*).

### Chemicals

Hexanoic (Cat No. H12137), octanoic (Cat No. O3907) and linoleic acids (Cat No. L1376), and denatonium benzoate (Cat No. D5765) were purchased from Sigma-Aldrich(St. Louis, MO) with a purity of >98%. Sucrose (Cat No. M1117) was purchased from Amresco (Solon, OH). Stock solutions of fatty acids were prepared in 80% ethanol, except for hexanoic acid (in 20% ethanol) and kept at −20° C. A stock solution for denatonium benzoate was prepared in Millipore Q water and kept at 4° C for up to one week. Stock solutions were diluted to the final concentration using Millipore Q water prior to each experiment. Sucrose and denatonium benzoate solutions were also mixed with ethanol to obtain the same final concentration of ethanol.

### Proboscis extension reflex (PER) assay

PER assays were carried out as described in Slone et al. (*Slone et al., 2007*) with some modifications. Briefly, files were collected on the day of eclosion and kept in standard corn meal food for 3–7 days at 25°C. Before performing PER assays, flies were starved for 24 to 26 hr at 25°C in vials with a water-saturated Whatman filter paper. Flies were immobilized by cooling on ice, mounted by their backs/wings on a microscope slide using double-sided Scotch tape and allowed to recover for 30 min at room temperature. Prior to the PER assay, flies were allowed to drink water until satiation to ensure PER responses were nutrient derived. Flies that showed no response to water were excluded. Taste solutions were delivered with a 10 μl pipette to legs for up to ~four s. Each fly was tested three times with each taste solutions, and flies were allowed to drink water between each new application. A PER response was recorded as positive (1) if the proboscis was fully extended, otherwise it was recorded as negative (0). PER response scores (%) from a single fly was 0% (0/3 responses in the three applications), 33% (1/3), 66% (2/3) or 100% (3/3).

### Immunofluorescence

Flies were aged for 4–7 days in standard corn meal food at 25°C before dissection. Labella were detached from the proboscis, while the tarsi were cut off above the fourth segment. Tissues were placed in eppendorf tubes containing fixation buffer (phosphate buffered saline with 4% paraformaldehyde and 0.2% triton X-100) for 30 min at room temperature. Tissues were washed twice in washing buffer (phosphate buffered saline containing 0.2% triton X-100) for 30 to 60 min at room temperature and incubated with the primary antibodies (chicken anti–GFP, 1:2000 dilution; rat anti–mCD8, 1:200 dilution, Thermo Fisher Scientific; rabbit anti-IR25a, 1:1000 dilution, (*Benton et al., 2009*); mouse anti-HA, 1:1000 dilution, Covance Research Product Inc.) at 4°C overnight in washing buffer containing 5% heat-inactivated goat serum. Tissues were washed twice in washing buffer for 30 to 60 min at room temperature and incubated with the secondary antibodies (goat anti–chicken ALEXA 488, 1:500 dilution, Thermo Fisher Scientific; goat anti–rat Cy3, 1:300 dilution; goat anti-rabbit Cy3, 1:300 dilution; goat anti-mouse Cy3, 1:100 dilution, Jackson Immunoresearch Laboratories Inc.) in washing buffer containing 5% heat-inactivated goat serum for 3 hr at room temperature. Tissues were washed twice in washing buffer for 30 to 60 min at room temperature. All incubations, except incubations with antibody, were performed under gentle mixing. Tissues were mounted with

VectaShield (Vector Lab) on a microscope slide and images were obtained using a Nikon A1R confocal microscope system.

## Calcium imaging

Calcium imaging for tarsal taste neurons was performed based on the protocol described in Miyamoto et al. (*Miyamoto et al., 2013*, *2012*) with minor modification. Briefly, female files were collected on the day of eclosion and kept in standard corn meal food for 3–8 days at 25°C. The prothoracic leg was cut with a razor blade between the femur and the tibia, and silicone lubricant (Dow Corning) was applied to seal the cut area of the leg. Legs were mounted laterally on a glass bottom culture dish (MatTek Corporation) using double-sided scotch tape and covered with 1% agarose, leaving the fourth and fifth tarsal segments exposed for ligand application. Millipore Q water (100 µl) was applied to cover the preparation, which was placed on the stage of a Nikon eclipse Ti inverted microscope. Images were obtained every 500 ms, starting 15 s before application and ending 105 s after ligand application. Three to five different ligands (100 µl) were tested in each preparation. Sucrose or denatonium ligands were included in each preparation to ensure functionality of sweet or bitter GRNs, respectively. The sequence of ligand application was sucrose (or denatonium) solution, carrier (0.2 or 1% ethanol in water) and fatty acid solution. When fatty acids were tested, 0.2% ethanol was used as a carrier for all $Ca^{2+}$ imaging experiments (1% ethanol when testing 2.5% hexanoic acid), because ethanol concentration above 1% destabilizes the preparation. Maximal concentration of octanoic and linoleic acids was 0.2% in 1% ethanol solution due to insolubility at higher concentrations. Preparations were washed with 200 µl of Millipore Q water five times after a ligand was tested, and equilibrated for 3 min before the next ligand was applied. Fluorescence light intensity of tarsal taste neurons was measured in the cell bodies. Background auto fluorescence, obtained from adjacent region, was subtracted. Baseline fluorescence was determined from the average of five frame measurements, taken immediately before ligand application. An equation for ΔF/F (%) was fluorescence light intensity of the cell body – baseline/baseline x 100. ΔF/F (max %) was calculated as the maximum value within 30 s after ligand application.

## Statistical analysis

Statistical analyses were conducted using Prism software (GraphPad Software 5.0 Inc.). PER assay and $Ca^{2+}$ imaging data were analyzed with nonparametric statistics because tested groups did not meet the assumption for normal distribution based on D'Agostino-Pearson and Shapiro-Wilk normality tests. For comparison between multiple groups (data of all figures, except *Figures 1B*, *4C*, *5B, C*, *6B*, *7D* and *Figure 1—figure supplement 1*, *Figure 4—figure supplement 1B* and *Figure 5—figure supplement 1B*), Kruskal-Wallis test by ranks (nonparametric one-way ANOVA) was performed to test for difference of rank distribution. As post hoc test, Dunn's multiple comparison tests were employed to compare two specific groups. Mann-Whitney U test (nonparametric t test) with two-tailed P value was used to compare means of two groups (*Figures 1B*, *4C*, *5B, C*, *6B*, *7D* and *Figure 1—figure supplement 1*, *Figure 4—figure supplement 1B* and *Figure 5—figure supplement 1B*). Sample size for PER assays and $Ca^{2+}$ imaging experiments were based on Slone et al. (*Slone et al., 2007*) and Miyamoto et al. (*Miyamoto et al., 2012*)

## Acknowledgements

We thank R Benton and C Montell for fly strains. This work was supported by grants 1RO1GMDC05606 and 1RO1DC13967 from National Institutes of Health.

## Additional information

### Funding

| Funder | Grant reference number | Author |
|---|---|---|
| National Institutes of Health | RO1GMDC05606 | Hubert O Amrein |
| National Institutes of Health | RO1DC13967 | Hubert O Amrein |

The funders had no role in study design, data collection and interpretation, or the decision to submit the work for publication.

## Author contributions
Ji-Eun Ahn, Data curation, Formal analysis, Investigation, Methodology; Yan Chen, Data curation, Yan Chen generated immunnostaining data for Figure 2 and Figure 2-figure supplement 1; Hubert Amrein, Conceptualization, Supervision, Funding acquistion, Investigation, Writing - original draft, Project administration, Writing - review and editing

## Author ORCIDs
Ji-Eun Ahn https://orcid.org/0000-0002-3390-3578
Yan Chen https://orcid.org/0000-0002-5321-3064
Hubert Amrein http://orcid.org/0000-0001-8799-7250

## Decision letter and Author response
Decision letter https://doi.org/10.7554/eLife.30115.033
Author response https://doi.org/10.7554/eLife.30115.034

## Additional files

### Supplementary files
• Transparent reporting form
DOI: https://doi.org/10.7554/eLife.30115.030

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
