## [Decision Letter]

Thank you for submitting your article "Molecular Basis of Fatty Acid Taste in *Drosophila*" for consideration by *eLife*. Your article has been favorably evaluated by a Senior Editor and three reviewers, one of whom, Mani Ramaswami (Reviewer #1), is a member of our Board of Reviewing Editors.

The reviewers have discussed the reviews with one another and the Reviewing Editor has drafted this letter to indicate our reservations about the depth of your analysis to support a quite interesting conclusion. We request that you respond to the essential revisions below..

Summary:

The paper identifies two IRs as components of a fatty acid receptor in sensory neurons, and shows that these are expressed in taste sensory neuron subclasses that respond via other receptors to sweet and bitter molecules respectively. Through several experiments and arguments, the authors propose that behavioural responses to fatty acids are mediated by higher level integration of signals from distinct groups of sensory neurons.

The detection of fatty acids is not only of fundamental interest but also relevant to host-seeking in disease vectors, which may use carboxylic acids as host-relevant cues. Thus, the findings are potentially significant. While, all the experiments provided are well done and convincing, the work does not at this stage make several key points as extensively or clearly as required for publication in *eLife*.

Essential revisions:

1) Most critically required is the clear identification of a receptor subunit that is specific to fatty acids. The work shows that the detection of the fatty acids hexanoic and octanoic acid by "sweet neurons" in the leg depends on and requires the Ionotropic receptors IR25a and IR76b. However, Ir76b and Ir25a, which are widely studied, are each broadly expressed and involved in detecting many different chemicals. The authors' observations add fatty acids to these chemicals, but given their wide expression, it seems unlikely that these two IRs constitute a receptor for fatty acids, at least without one additional, specifically expressed IR involved. So, while the findings are novel, they do not yet describe molecular basis of fatty acid detection yet. At this point, implicating ir25a and ir76b in sensing a given chemical is not enough. A revised manuscript must make substantial progress towards discovering the specificity components.

2) Since NorpA (a phospholipase C) is required for the behavioral response, the authors should examine whether the responses they see in the sweet leg neurons are affected in the NorpA mutants. Either way this will be an important result. NorpA could act independently of the IRs or, alternatively, upstream or downstream of the IRs. In the latter case, it would be quite exciting as mammalian fatty acid detection involves GPCRs and such an observation would indicate that the IRs are acting permissively, downstream of such receptors. Viable *norpA* nulls are available (it is even on the X making imaging in the mutant background easier), so this should be a relatively simple experiment.

3) It is important to know whether Ir8a is involved in the response, as a recent study using Anopheles IRs (Pitts et al., 2017) shows a different IR combination, IR8a/IR75k, forms a hexanoic/octanoic acid receptor in oocytes. The expression data suggests Ir8a probably isn't involved, but since there are Ir8a nulls available to formally test this, it should be examined. This is also a simple experiment.

4) The data is based on detection by the tarsi. The fact that this does not apply to the labellum is cited as data not shown, and is not explored further. The Discussion reads "Moreover, labellar stimulation using fatty acids induces weak if any PER responses (J.-E.A. and H.A., unpublished data)." And further down, "Intriguingly, these non/chemosensory IR complexes share a common theme with the fatty acid receptors, and carboxylic acid receptors that mediate sour taste in the non/bitter/non-sweet GRN of tarsal taste sensilla (Y.C. and H.A., unpublished data)…" Relevant data should be shown in the Results section and discussed clearly.

5) In Figure 3, data are shown for Ir25a (C,D), but not for 76b. As all others were carried out in tandem for the two. Data should be shown for both or there should be an explanation for the omission.

6) The mechanism by which sweet and bitter neurons might interact is touched upon but not developed. The authors state "While the subset of tarsal bitter GRNs activated by hexanoic acid does not require either gene, we have discovered that the third GRN (the sour GRN) is narrowly tuned to acids in a IR25a/IR76b dependent manner (Y.C. and H.A., unpublished data)." Intriguing, but not helpful for the manuscript.

7) One could also take Figure 6 bit more in a rigorous direction. For example, there is only one concentration (2.5% hexanoic acid) for the critical point that the paper wants to make. What about mixtures (the authors mentioned the differences in solubility in water, but precisely this point could perhaps be addressed?). It also seems somewhat anticlimactic to end with Figure 6. These suggest a feeling of incompleteness. Also, the response is not general to carboxylic acids, but just hexanoic acid, so these GNs may not be important regulators of overall fatty acid intake.

With strong and clear responses to the issue of how fatty acids are detected by sweet sensory neurons (and by testing whether the specificity factor is also expressed in bitter/ sour GRNs) it is possible that points 6 and 7 could be addressed by rewriting the manuscript to more clearly indicate what is known and hypothesised about FA sensing via other GN classes.

---

## [Author Response]

Essential revisions:1) Most critically required is the clear identification of a receptor subunit that is specific to fatty acids. The work shows that the detection of the fatty acids hexanoic and octanoic acid by "sweet neurons" in the leg depends on and requires the Ionotropic receptors IR25a and IR76b. However, Ir76b and Ir25a, which are widely studied, are each broadly expressed and involved in detecting many different chemicals. The authors' observations add fatty acids to these chemicals, but given their wide expression, it seems unlikely that these two IRs constitute a receptor for fatty acids, at least without one additional, specifically expressed IR involved. So, while the findings are novel, they do not yet describe molecular basis of fatty acid detection yet. At this point, implicating ir25a and ir76b in sensing a given chemical is not enough. A revised manuscript must make substantial progress towards discovering the specificity components.

We have addressed this concern and reported the identification of a third subunit necessary for fatty acid taste in sweet GRNs, presented in Figure 5. In brief, we carried out PER analysis on flies in which the function of IR gene candidates was disrupted my mutation or RNAi in sweet GRNs. Of seven candidate genes, knock down of one – IR56d – caused a severe reduction of PER responses to fatty acids.

Expression analysis using *IR56d-GAL4* line revealed that this gene is co-expressed in sweet GRNs of tarsi. Importantly, knockdown of *IR56d* in sweet neurons led to almost complete loss of fatty acid induced activity, while no effect on sugar responses was observed. Thus, we found a third sweet GRN specific subunit that appears to provide fatty acid specificity.

2) Since NorpA (a phospholipase C) is required for the behavioral response, the authors should examine whether the responses they see in the sweet leg neurons are affected in the NorpA mutants. Either way this will be an important result. NorpA could act independently of the IRs or, alternatively, upstream or downstream of the IRs. In the latter case, it would be quite exciting as mammalian fatty acid detection involves GPCRs and such an observation would indicate that the IRs are acting permissively, downstream of such receptors. Viable norpA nulls are available (it is even on the X making imaging in the mutant background easier), so this should be a relatively simple experiment.

We performed Ca^2+^ imaging experiments with fatty acids on tarsal sweet GRNs in *norpA* mutant flies. Indeed, *norpA* mutant flies show a severe reduction of Ca^2+^ responses to fatty acids, but not sugar. This indicates that both IRs as well as phosopholipase C act upstream of the Ca^2+^ channel that ultimately activates the GRNs. This observation suggests that phospholipase C plays a somewhat novel role in neuronal signaling (see discussion). However, because this is somewhat accessory finding to the main thrust of the paper, we included these results as a figure supplement (Figure 1—figure supplement 2).

3) It is important to know whether Ir8a is involved in the response, as a recent study using Anopheles IRs (Pitts et al., 2017) shows a different IR combination, IR8a/IR75k, forms a hexanoic/octanoic acid receptor in oocytes. The expression data suggests Ir8a probably isn't involved, but since there are Ir8a nulls available to formally test this, it should be examined. This is also a simple experiment.

*IR8a* mutants (*IR8a^1^*) have normal PER responses to fatty acids and sugars, which was expected (Author response image 1). The data is appended for the perusal of the reviewers, but we do comment on it in the text, as there is clear evidence that either IR25a or IR8a (but not both) participate in IR channels from work by the Benton lab. Moreover, there is evidence that IR8 is not expressed in the taste system (Benton lab, Cameron et al., 2010, and our own unpublished data).

**Author response image 1. respfig1:** PER responses of *IR8a* mutant flies (*IR8a^1^*) to fatty acids. Each bar represents the mean ± SEM of PER responses (n = 35- 55 flies). NS indicates a no significant difference between *IR8a* mutant and control (*w^1118^*) flies (Two-tailed, Mann-Whitney U test, p < 0.05, ns: not significant). Each y-axis delineates groups for Mann-Whitney U test.

4) The data is based on detection by the tarsi. The fact that this does not apply to the labellum is cited as data not shown, and is not explored further. The Discussion reads "Moreover, labellar stimulation using fatty acids induces weak if any PER responses (J.-E.A. and H.A., unpublished data)." And further down, "Intriguingly, these non/chemosensory IR complexes share a common theme with the fatty acid receptors, and carboxylic acid receptors that mediate sour taste in the non/bitter/non-sweet GRN of tarsal taste sensilla (Y.C. and H.A., unpublished data)…" Relevant data should be shown in the Results section and discussed clearly.

We included PER analysis on labellar taste (Figure 1—figure supplement 1). In our hands, PER responses to fatty acid are more robust after stimulation of tarsi, than stimulation of the labellum. Nevertheless, both organs respond to fatty acids with PER. Because Ca^2+^ imaging at the cellular level is only possible at this time from tarsal GRN, and all our imaging was done there, we show PER data from tarsi in the main figure.

In addition, the data cited as “Y.C. and H.A., unpublished data” is now published (Chen and Amrein, 2017) and cited appropriately.

5) In Figure 3, data are shown for Ir25a (C,D), but not for 76b. As all others were carried out in tandem for the two. Data should be shown for both or there should be an explanation for the omission.

We appreciate that reviewers pointed out the lack of explanation why rescue/RNAi was done for IR25a but not IR76b. Our rationale was that we wanted to correlate – as good as possible – cellular responses of the sweet neurons with basic appetitive taste responses (i.e. PER), and choosing IR25a as the core subunit made the most sense. In any case, we now show that PER responses are rescued when *IR76b* mutants express *UAS-IR76b* in sweet neurons only (shown Figure 3). We do not have an RNAi line for *IR76b*. Regardless, rescues in sweet GRNs in both mutant strains show that sweet GRNs mediate PER responses to fatty acids. Note that we removed the RNAi data in the revised manuscript due to redundancy with the more conclusive gene knock-outs and rescue experiments.

7) One could also take Figure 6 bit more in a rigorous direction. For example, there is only one concentration (2.5% hexanoic acid) for the critical point that the paper wants to make. What about mixtures (the authors mentioned the differences in solubility in water, but precisely this point could perhaps be addressed?). It also seems somewhat anticlimactic to end with Figure 6. These suggest a feeling of incompleteness. Also, the response is not general to carboxylic acids, but just hexanoic acid, so these GNs may not be important regulators of overall fatty acid intake.

We are uncertain exactly what the criticism is, but we would clarify questions regarding concentrations, and why this experiment was carried out with hexanoic acid, but not the other two acids. First, fatty acids are notoriously difficult to work with, much more so than water-soluble sugars and amino acids obviously, but also as bitter compounds (which activate neurons at much lower concentrations).

Because fatty acids are food compounds, they activate neurons only at much higher concentrations (compared to bitter compounds), and it is challenging to solubilize them at the necessary concentration with solvents that maintain clarity of the solution and integrity of GRNs.

We cannot exclude the possibility that activation of bitter GRNs by high concentration of hexanoic acids might not be a general feature for all fatty acids (stated in the manuscript). Nevertheless, our data provides a cellular explanation for the observed behavior to hexaonoic acid. Thus, we think this is an important result, because taste cues activating multiple (and opposing) GRNs have not been described widely, at least not in *Drosophila*. This observation could also provide a mechanism that allows discrimination between sweet and fatty acid chemicals.